# Audit as a Tool for Improving the Quality of Stroke Care: A Review

**DOI:** 10.3390/ijerph20054490

**Published:** 2023-03-03

**Authors:** Irene Cappadona, Francesco Corallo, Davide Cardile, Augusto Ielo, Placido Bramanti, Viviana Lo Buono, Rosella Ciurleo, Giangaetano D’Aleo, Maria Cristina De Cola

**Affiliations:** IRCCS Centro Neurolesi Bonino-Pulejo, S.S. 113 Via Palermo, C.da Casazza, 98124 Messina, Italy

**Keywords:** audit, feedback, stroke, rehabilitation, prevention, EASY-NET

## Abstract

Introduction: A clinical audit is a tool that allows the evaluation of and improvement in the quality of stroke care processes. Fast, high-quality care and preventive interventions can reduce the negative impact of stroke. Objective: This review was conducted on studies investigating the effectiveness of clinical audits to improve the quality of stroke rehabilitation and stroke prevention. Method: We reviewed clinical trials involving stroke patients. Our search was performed on PubMed databases, Web of Science, and Cochrane library databases. Of the 2543 initial studies, 10 studies met the inclusion criteria. Results: Studies showed that an audit brought an improvement in rehabilitation processes when it included a team of experts, an active training phase with facilitators, and short-term feedback. In contrast, studies looking at an audit in stroke prevention showed contradictory results. Conclusions: A clinical audit highlights any deviations from clinical best practices in order to identify the causes of inefficient procedures so that changes can be implemented to improve the care system. In the rehabilitation phase, the audit is effective for improving the quality of care processes.

## 1. Introduction

Stroke is one of the main causes of death and disability worldwide, causing 5 million deaths [1]. According to the World Health Organization (WHO), an instance of ictus occurs every 5 s [2]. Stroke is a clinical syndrome associated with rapidly developing signs of focal or global loss of cerebral functions, with a cause of vascular origin [3].

The acute phase is extremely important for a successful rehabilitation; in fact, there is a therapeutic window during which intervention is more likely to modify the course of the disease and successfully lead to neuronal reactivation [4,5]. Receiving organized hospital care in a stroke unit is associated with patients being more likely to be alive, independent, and living at home 1 year after their stroke compared to patients who do not receive such specialized care [6]. Preventive interventions are also essential to reduce the risk of recurrence. Prevention processes include encouraging a healthy lifestyle with regular physical activity and balanced nutrition to keep body weight and blood cholesterol levels under control, and with limits on alcohol, smoking and drug consumption [7]. 

For the treatment of stroke, therefore, phases of rehabilitation and preventive care are extremely important. For this reason, it is critical to find ways to evaluate and improve stroke care processes. An adequate tool to evaluate these elements is the audit. 

The clinical audit is seen as one approach to improving the quality of patient care [8,9]. Specifically, a clinical audit is the systematic, critical analysis of the quality of medical care, including the procedures used for diagnosis and treatment, the use of resources, and the resulting outcome and quality of life for the patient. In other words, the audit is the process of reviewing the delivery of care to identify deficiencies so that they may be remedied [10,11]. 

The clinical audit can be described in four main phases: (i) planning (stating the aim, defining improvement, deciding quantifiable change); (ii) doing (collecting data, monitoring progress, providing feedback); (iii) studying (discussing data, assessing data, interpreting data); and (iv) acting (continued action) [12]. Clinical audits are largely used in medical care, both locally (local hospitals and medical centers) and nationwide (to improve the national health system). However, since audits are rarely published and available to the wider community, it is hard to both identify a common practice and evaluate their outcomes [13]. 

Indeed, no agreement exists about which audit methodologies are the most suitable approach, and, not surprisingly, there is significant confusion among healthcare professionals about how to implement an audit and integrate it effectively into clinical practice [10]. 

Given the high variability in audit methodologies, and the importance of improving clinical practice for stroke care, this review focused on the studies that investigated the efficiency of clinical audits to improve quality care for stroke (rehabilitation and prevention) by taking into account clinical trials carried out on patients with ictus.

## 2. Materials and Methods

A descriptive review was conducted on studies that performed a trial on the care process for rehabilitation and prevention of stroke that used the audit for assessment of quality care. 

Studies were identified by searching PubMed (from 1972 to 2022), Web of Science (from 1991 to 2022), and the Cochrane library (from 1989 to 2022) databases, published before 7 October 2022. The keyword search was conducted by one researcher and took about 2 days. 

The search keywords were “stroke” AND “audit” (“stroke” [MeSH Terms] OR “stroke” [All Fields]) AND (“audit” [MeSH Terms] OR “quality of care” [All Fields] OR “assessment of quality” [All Fields] OR “improvement of care” [All Fields] OR “improvement of quality” [All Fields] OR “revision process” [All Fields] OR “revision of care” [All Fields]). After duplicates had been removed, all articles were evaluated based on title, abstract, and text.

Studies that met the following criteria were included in this review:-Studies of audits; -Prevention and rehabilitation of stroke;-Clinical trials;-Studies in English;-No review; -There were no restrictions regarding the year of publication (details about the year of publication found in each database search are specified above).

Evaluation of the studies was completed over three rounds (each study was tripled-checked for inclusion) by the same researcher who carried out the keyword search. This phase took about 1 month.

## 3. Results

Out of the initial 2543 studies identified from our search, 10 studies met the inclusion criteria (Figure 1). 

All included studies examined the quality of stroke care using the audit as an assessment tool (see Table 1). Of 10 studies that evaluated the quality of stroke care, 7 regarded rehabilitation [14,15,16,17,18,19,20] and 3 regarded prevention [21,22,23]. For a detailed description of different audit interventions see Table 2. 

### 3.1. Studies on Rehabilitation

Of the seven studies that evaluated the quality of rehabilitation in stroke patients, five audited both intervention and control groups [14,15,18,19,20], while two studies audited the intervention group only [16,17]. 

Overall, results from studies that audited both groups show that an audit is an effective tool to improve the quality of rehabilitation in stroke patients. For example, Power et al., 2014 [14] reported a 10.9% improvement after implementing a Breakthrough Series (BTS) intervention, which includes a team of quality improvement experts, three training meetings, and an implementation phase. McGillivray et al., 2017 [18] and Hinchey et al., 2010 [19] found encouraging results for audit effectiveness, specifically when feedback was given within one day by a coordinating nurse (concurrent review) and when a multifaceted intervention was included. Consistent with these studies, Joliffe et al., 2020 [20] showed audit-related improvement specifically when therapists received the facilitator-mediated guideline package. Sulch et al., 2002 [15] carried out a study on 152 patients comparing whether the Integrated Care Pathway (ICP)—i.e., a multidisciplinary set of progressive care delivered within a specific time frame—improves the quality of care compared to routine care. Results showed that ICP, compared to routine care, was associated with greater improvement in initial assessments, better documentation of the diagnosis, and a higher rate of discharge within 24 h.

In contrast to the studies just presented that audited both groups, Linch et al., 2016 [16] and Machine-Carrion et al., 2019 [17] carried out an investigation of the effectiveness of the multifaceted intervention by auditing the intervention group only.

In particular, Lynch et al., 2016 [16] carried out a study in Australia involving a total of 586 patients over a period of 14 months. The objective was to compare two groups, one receiving education-only intervention and one receiving a multifaceted intervention, with the audit performed on the latter group only. The results showed that, similarly across the two groups, the odds for a patient to receive an assessment for rehabilitation were 3.69 times greater in the post-intervention period compared to the pre-intervention period, with no difference between the two interventions. 

In the study by Machine-Carrion et al., 2019 [17], patients from hospitals that had received usual care (no intervention) were compared with patients in hospitals that had received the multifaceted intervention, on which the audit was performed. Patients in intervention hospitals were more likely to receive all acute therapies during hospitalization than those in control hospitals.

### 3.2. Studies on Prevention

Of the three studies evaluating stroke prevention, two audited both experimental groups (i.e., intervention and control) [21,23], while one audited the intervention group only [22].

Wright et al., 2007 [23] carried out a study of approximately 2800 patients in the UK, finding an improvement in patients’ adherence to atrial fibrillation and TIA therapy, which was significantly greater in the intervention group—who attended 5 meetings to improve adherence within guidelines—compared to the control group.

With a similar sample, Williams et al., 2016 [22] compared 12 hospitals in the USA and 2164 patients to see if a training intervention plus indicator feedback was more effective for improving quality than indicator feedback alone. The training intervention plus indicator feedback was associated with improvement in venous thrombosis prophylaxis (DVT), but the effect was not sustained long-term.

With a much larger sample of 12,766 patients, Geary et al., 2019 [21] carried out a study in Sweden, with the aim of improving the diagnostics and use of preventive drugs for stroke. In diagnosing TIA, but not ischemic stroke, there was an improvement in the intervention group compared with the control group. Instead, regarding preventive drugs use, the audit and feedback intervention did not lead to any improvement in patients with ischemic stroke/TIA. 

## 4. Discussion

Stroke can be life-threatening in the short term and can cause a reduction in quality of life and, consequently, physical, emotional, and behavioral disabilities [6,24]. Over the past two decades, A&F strategies have been used for all areas of health care, namely, preventive, acute, chronic, and palliative care, with the aim of improving the quality of performance, reducing errors, and increasing safety [25]. 

First, considering stroke rehabilitation, the studies included here showed that an audit was generally associated with improved care processes. Specifically, the audit of rehabilitation interventions brought further improvements when it included a team of experts [14,15,19], an active training phase with facilitators [14,18,20], and concurrent, short-term feedback [18].

These conclusions agree with what Welsh et al., 1993 [26] identified as key factors for audit success. These include an enabling organizational environment, strong leadership and direction of audit programs, strategy and planning in audit programs, resources and support for audit programs, monitoring and reporting of audit activity, commitment and participation, and high levels of audit activity which is seen by its participants as engaging and relevant, with respects to its nature and impact. More recent studies further suggest that audit success increases if the audit and feedback is provided by professionals who are admired by healthcare professionals [27,28]. Furthermore, it has been demonstrated that it is essential for audit success that personnel trust the data being investigated and consider the clinical topics being audited important [29,30]. A prerequisite for an optimal audit leading to change is that clinicians are committed to behavior change [31].

In contrast, an audit of stroke prevention interventions seems to report conflicting results, and studies are too few to reach satisfactory conclusions. In particular, of the three studies examined, while two reported audit-related improvements in stroke prevention care, these results are limited by small samples [22,23] and lack of assessment of long-term effects [22]. When a much larger sample was considered, there were no positive effects of the audit on the prevention of ischemic stroke [21].

It was possible that some of the main barriers to clinical audit identified by Robinson 1996 [32] may have been responsible for the null results found in our review in relation to prevention care, including a lack of resources, lack of expertise or advice in project design and analysis, relationships between groups and group members, lack of an overall plan for audit, and organizational impediments. 

Additional factors of audit failure could be due to lack of clear and easy-to-understand feedback [33,34] and a lack of cooperation and motivation from the parts involved [35]. Springer et al., 2021 [35], in fact, highlighted that healthcare professionals working as a team during the audit and feedback process have improved stroke care. However, it remains unclear why these barriers would have affected the success of the clinical audit on prevention care specifically, while rehabilitation care was found to be improved overall, at least across the studies reviewed here. It may be that the clinical audit may be a tool for improving patient care differently depending on whether care applies to prevention or rehabilitation.

Furthermore, the studies reviewed found that audits were performed differently, possibly due to variability in healthcare workforce knowledge of clinical audits [36] and/or variability in access to funds [10], making a systematic comparison of the various protocols difficult (Table 2) [10,37].

This review included a small number of papers as only 10 studies met the inclusion criteria. This, combined with the significant variability in methodologies among the cited studies, which in turn led to significant variability in study results, including both quantitative and qualitative outcomes, did not allow objective comparisons to be made between investigations. This made it impossible to conduct a meta-analysis, making it difficult to reach satisfactory conclusions; further studies are needed to verify the effectiveness of audits as a tool to improve interventions for stroke prevention. Despite these limitations, the present review is, to the best of our knowledge, the first to provide a detailed picture of the clinical trials that have evaluated the usefulness of audits in improving the quality of care for stroke patients, allowing for a systematic evaluation of audit effectiveness and identification of weaknesses, in turn enabling improvement so that more efficient future studies can be designed.

## 5. Conclusions 

In conclusion, in light of the reviewed studies, the audit appears to be effective in improving the quality of care for stroke patients in the rehabilitation phase. More studies are needed to reach robust conclusions with regard to the preventive phase. Future studies should focus on applying standardized audit protocols to advance improvements in stroke care and allow for systematic comparisons between studies [38]. In order to improve clinical practice, a number of references may provide appropriate learning sessions and educational material regarding the following: the theory and practice of improvement [14,17,22]; meetings to improve adherence to guidelines [23,39]; case reviews conducted by researchers trained in the National Stroke Audit methodology [15]; regular conferences on quality measures and discussions of aspects that need to be improved [17]; and systems capable of quickly and regularly giving feedback [18].

## Figures and Tables

**Figure 1 ijerph-20-04490-f001:**
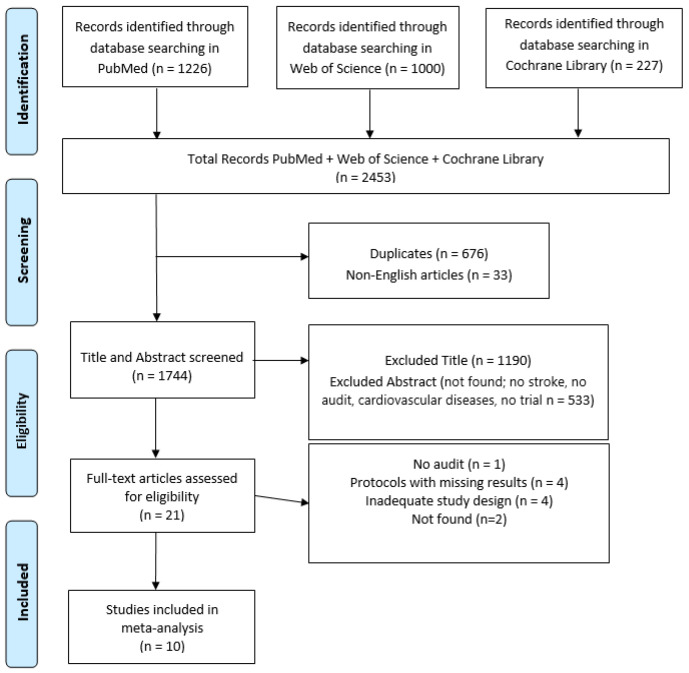
Graphic of identification, screening, eligibility, and inclusion of review article.

**Table 1 ijerph-20-04490-t001:** Studies’ characteristics.

References	Aim	Sample (N)	Country	Groups	Type of Intervention	Duration	Outcomes
Power et al., 2014 [14]	Improve patient compliance and reliability via Breakthrough Series (BTS) model	24 hospitals 360 patients	UK	group 1: control; group 2: intervention Qic (BTS)	rehabilitation	18 months	10.9% improvement compared to the control group
Sulch et al., 2002 [15]	Compare Integrated Care Pathway (ICP) based management and routine (standard) multidisciplinary care.	152 patients	UK	group 1: traditional multidisciplinary intervention; group 2: integrated care pathway	rehabilitation	-	(1) a greater number of patients receiving initial assessments and investigations; (2) better documentation of discussions on diagnosis, prognosis, and follow-up arrangements with patients and carers; (3) a significantly higher proportion of discharge notification within 24 h.
Lynch EA., et al., 2016 [16]	Compare education-only and multifaceted interventions for improving rehabilitation assessment	10 hospitals 586 patients	Australia	group 1: education-only group 2: multifaceted intervention	rehabilitation	14 months	3.69 times more likely to receive rehabilitation compared to pre-intervention, with no difference between the two intervention groups.
Machline-Carrion MJ et al., 2019 [17]	Assess the effect of a multifaceted quality improvement intervention on adherence to evidence-based therapies	36 hospitals 1624 patients	Brazil, Argentina, Peru	group 1: usual care group 2: multifaceted strategy	rehabilitation	17 months	Patients in the intervention hospitals were more likely to receive all acute therapies during hospitalization.
Geary et al., 2019 [21]	Improve the use of secondary preventive stroke medications and diagnosis recording in ischemic stroke/TIA	12,766 patients	Sweden	group1: control; group 2: audit and feedback	prevention	18 months	Neutral results for secondary preventive medications in ischemic stroke/TIA patients. Diagnosis of TIA, but not of ischemic stroke, was better in intervention PCCs after the intervention.
McGillivray et al., 2017 [18]	Evaluate whether concurrent review implementation was associated with change in performance on stroke measure outcome data	2 hospitals, 620 medical reports	USA	group 1: control; group 2: concurrent review	rehabilitation	100 months	Statistically significant improvements in 8 of the 10 stroke measures in hospitals that implemented concurrent reviews opposed to control hospitals.
Williams et al., 2016 [22]	Compare quality improvement training plus indicator feedback and indicator feedback alone.	12 hospitals 2164 patients	USA	group 1: indicator feedback; group 2: quality improvement training plus indicator feedback	prevention	36 months	Quality improvement training was associated with early DVT improvement, but the effect was not sustained over time and was not seen with dysphagia screening.
Hinchey et al., 2010 [19]	Assess a multifaceted intervention designed to improve adherence to guideline-recommended processes	13 hospitals 3311 patients	USA	group 1: (control group) received audit, feedback, and benchmark information; group 2: (intervention group) group 1 plus a multifaceted intervention	rehabilitation	40 months	Multifaceted QI intervention was associated with improvement in appropriate anticoagulation on discharge compared to audit, feedback, and benchmarking alone.
Joliffe et al., 2020 [20]	Compare two implementation packages on guideline adherence	55 patients 29 therapist	Australia	group 1: usual care; group 2: facilitator-mediated package; group 3: self-directed implementation package	rehabilitation	3 months	Improvements in guideline adherence by therapists who received the facilitator-mediated package, inclusive of multiple implementation strategies.
Wright et al., 2007 [23]	Evaluate clinical effectiveness of implementing evidence-based guidelines	63 hospitals 2896 patients	UK	group 1: guidelines for TIA; group 2: guidelines for AF	prevention	missing	Compliance was significantly greater in the condition for which practices had received the implementation program.

**Table 2 ijerph-20-04490-t002:** Type of audit.

References	Audit Sample	Audit Recipients (Test Group; Control Group; Both)	Measurement Methodology	Audit Intervention
Power M. et al., 2014 [14]	360 patient	both groups	Model BTS (Breakthrough Series)	Learning sessions on improvement theory and practice; Implementation of the BTS model; Data collection on 20 randomly selected patients per month; Sending data to a customized web-based system linked to a national audit.
Sulch et al., 2002 [15]	152 patients	both groups	Intercollegiate Stroke Audit	Analysis of care processes; Case review conducted by researchers trained in National Stroke Audit methodology but blinded; Main outcome measures: Percentage of patients receiving recommended interventions; Secondary outcome measures: compliance with the ICP and time taken to carry out the interventions on schedule.
Lynch EA., et al., 2016 [16]	303 patients	experimental group	multifaceted intervention	Control of medical records and subsequent verbal and written feedback; Workshop to identify barriers and develop strategies to support the use of art; Verbal feedback of the workshop and action plans to all participants; Samples were contacted by email or phone in the month following the surgery; A period of 4 months was selected for the sites to implement all the strategies for the implementation of the art and the improvement of assessment and rehabilitation practices.
Machline-Carrion MJ et al., 2019 [17]	844 patients	experimental group	multifaceted intervention	Case management; Sending reminders; Control of the therapeutic plan; Creation of didactic material; Audit and performance feedback; Regular conferences on quality measures and discussions of the aspects to improve.
Geary et al., 2019 [21]	6408 patients	experimental group	structured, healthcare database-derived quality reports on secondary preventive medication use and diagnosis recording	Preparation of center-specific quality reports; Report on data collection of all healthcare consumption; Sending an intervention via email and postal mail to the director of each center inviting the directors to disseminate the quality reports to the physicians of their center; Establishment of an e-mail box to provide an opportunity to ask questions about the material.
McGillivray et al., 2017 [18]	620 medical records	both groups	simultaneous review	Daily examination of stroke patients; Evaluation of physician care; Direct feedback provided by a coordinating nurse within one day; Performance monitoring and saving of information on an excel sheet;
Williams et al., 2016 [22]	2164 patients	both groups	missing	Monthly and quarterly feedback to hospitals on the performance of the two core and nine additional stroke quality indicators; The control sites have not received external training or facilitation;
Hinchey et al., 2010 [19]	3311 patients	both groups	Stroke Practice Improvment Network (SPIN)	Surveys related to physicians’ knowledge and attitudes, existing organizational infrastructures, and barriers to care delivery; Data collection and results analysis; Analysis of obstacles and identification of resources for improvement;
Jolieffe et al., 2020 [20]	55 patients	both groups	Theoretical Domains Framework Behavior Change Wheel	Assessment of therapists’ adherence to guidelines through audits and feedback of medical records (before and after the intervention); Evaluation of upper limb outcomes before and after surgery; Collection of self-reported weekly therapy minutes (patient reported and therapist reported)
Wright et al., 2007 [23]	2896 patients	both groups	missing	Meetings (No. 5) to develop adherence to evidence-based guidelines for the management of patients with atrial fibrillation; Meetings held to improve the quality of care using the guidelines, identify obstacles and incentives for change, agree on strategies appropriate to the situation; Step approach to the guidelines divided into 3 phases: phase 1 (training meetings), phase 2 (educational awareness visit), phase 3 (mail dissemination and reinforcement interventions).

## Data Availability

Data sharing is not applicable to this article as no new data were created or analyzed in this study.

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
