# Peer review of "Audit as a Tool for Improving the Quality of Stroke Care: A Review"

_ijerph, 2023, doi:10.3390/ijerph20054490_

Round 1

Reviewer 1 Report

Review of a manuscript -Manuscript ID: ijerph- 2225401

The topic of the work is interesting and encourages you to read the content of the review.

I see the following shortcomings in my work:

Material and methods section - who checked, how many people, and how many times were not specified.

Figures are signed under, not above, the figure.

I see translation errors in the tables, or omissions of translation into English, e.g.: "gruppo sperimentale"?

Otherwise, the manuscript is well organized, and the methods are well described.

It is a pity, however, that this is not a systematic review registered in PROSPERO. I suggest authors consider this. This would involve making adjustments, but it would certainly increase the power of the study, and thus make it more attractive to readers.

The work requires corrections before being allowed for publication.

Author Response

Material and methods section - who checked, how many people, and how many times were not specified.

Done. We included details as requested

Figures are signed under, not above, the figure.

Done. We have changed the placement of the title below the figure

I see translation errors in the tables, or omissions of translation into English, e.g.: "gruppo sperimentale"?

Done. We have corrected translation errors in the manuscript.

 Otherwise, the manuscript is well organized, and the methods are well described.

It is a pity, however, that this is not a systematic review registered in PROSPERO. I suggest authors consider this. This would involve making adjustments, but it would certainly increase the power of the study, and thus make it more attractive to readers.

Thanks for the suggestion. Unfortunately, PROSPERO only allows registration before study screening begins. We will take this into consideration for future works.

The work requires corrections before being allowed for publication.

Done.

Reviewer 2 Report

I would to thanks all team share in this work ,great effort but need more editing consideration 

Author Response

I would to thanks all team share in this work, great effort but need more editing consideration.

We thank you for your comment. We have edited the manuscript according to your suggestion

Reviewer 3 Report

Dear authors,

The issue of auditing stroke treatment and rehabilitation is actually a very unique issue. I'm not sure if all the articles on this topic in the literature have been scanned. It doesn't look like a carefully written review. Meta-analysis of the articles included in the review was not performed. The results related to these are not disclosed and there are no informative materials such as tables and graphs. They should also include other assemblies that contain the stroke related control and write them in the discussion section

Author Response

The issue of auditing stroke treatment and rehabilitation is actually a very unique issue. I'm not sure if all the articles on this topic in the literature have been scanned. It doesn't look like a carefully written review. Meta-analysis of the articles included in the review was not performed. The results related to these are not disclosed and there are no informative materials such as tables and graphs. They should also include other assemblies that contain the stroke related control and write them in the discussion section.

Thanks for the comments. We reassure the reviewer that all studies resulting from the keyword search were scanned according to the inclusion criteria and checked multiple times. We've also updated the methods to include more details. With regards to the meta-analysis, we have further explained in the text why it was not possible to perform it. We have also expanded the discussions with more recent literature.

Reviewer 4 Report

Dear Author/s

Thanks for your applied and valuable manuscript in term of review article.

The title was good but it seems that using "improving" instead of "assess" was much better but that was sound and clear enough.

The abstract was concise and precise and well structured.

The introduction was comprehensive and helpful.

Method was well defined with good search strategy in form of Figure.

Results were pretty good in the format of table including studies characteristics and table including type of audit.

Discussion was good and the comparison between rehabilitation and prevention in the matter of stroke was well described.

The conclusion was good and related and consistent with result & discussion.

References were relevant and partially new according to subject.

Be successful 

Regard

Author Response

Dear Author/s

Thanks for your applied and valuable manuscript in term of review article.

The title was good but it seems that using "improving" instead of "assess" was much better but that was sound and clear enough.

Done

The abstract was concise and precise and well structured.

The introduction was comprehensive and helpful.

Method was well defined with good search strategy in form of Figure.

Results were pretty good in the format of table including studies characteristics and table including type of audit.

Discussion was good and the comparison between rehabilitation and prevention in the matter of stroke was well described.

The conclusion was good and related and consistent with result & discussion.

References were relevant and partially new according to subject.

Be successful 
